# GAPSO-Optimized Fuzzy PID Controller for Electric-Driven Seeding

**DOI:** 10.3390/s22176678

**Published:** 2022-09-03

**Authors:** Song Wang, Bin Zhao, Shujuan Yi, Zheng Zhou, Xue Zhao

**Affiliations:** 1College of Engineering, Heilongjiang Bayi Agricultural University, Daqing 163319, China; 2College of Software, Shanxi Agricultural University, Taigu, Jinzhong 030801, China

**Keywords:** electric-driven seeding, genetic particle swarm optimization, fuzzy PID, control strategy

## Abstract

To improve the seeding motor control performance of electric-driven seeding (EDS), a genetic particle swarm optimization (GAPSO)-optimized fuzzy PID control strategy for electric-driven seeding was designed. Since the parameters of the fuzzy controller were difficult to determine, two quantization factors were applied to the input of the fuzzy controller, and three scaling factors were introduced into the output of fuzzy controller. Genetic algorithm (GA) and particle swarm optimization (PSO) were combined into GAPSO by a genetic screening method. GAPSO was introduced to optimize the initial values of the two quantization factors, three scaling factors, and three characteristic functions before updating. The simulation results showed that the maximum overshoot of the GAPSO-based fuzzy PID controller system was 0.071%, settling time was 0.408 s, and steady-state error was 3.0693 × 10^−5^, which indicated the excellent control performance of the proposed strategy. Results of the field experiment showed that the EDS had better performance than the ground wheel chain sprocket seeding (GCSS). With a seeder operating speed of 6km/h, the average qualified index (*I_q_*) was 95.83%, the average multiple index (*I_mult_*) was 1.11%, the average missing index (*I_miss_*) was 3.23%, and the average precision index (*I_p_*) was 14.64%. The research results provide a reference for the parameter tuning mode of the fuzzy PID controller for EDS.

## 1. Introduction

The quality of seeding greatly depends on the rotating speed of the metering disc driven by the seeding motor. Motor control strategies mainly include PID control, adaptive control, vector control, and intelligent control. The intelligent control has a blocked hierarchical control organization structure, which is convenient for handling a large amount of information, stored knowledge ability, and reasoning ability. Intelligent control carried out real-time control of the dynamic environment of the seeding motor movement process according to the characteristics of nonlinear, time-varying, dynamic, and complex uncertain control objects under the steady state, when driving the seeding motor speed regulation in the electric-driven seeding (EDS) [1,2,3]. Intelligent control strategies mainly include: fuzzy control [4,5,6,7,8], neural network [9,10,11,12], and the heuristic optimization algorithm [13,14,15,16,17]. Among them, the heuristic optimization algorithm has a significant role in tuning the parameters of the controller. The stronger the global optimization ability of the algorithm, the better the robustness of the controller adjusted by the algorithm. In recent years, scholars in various fields have also improved and applied the heuristic optimization algorithm and achieved remarkable achievements [18,19,20].

Zhai et al. [21] designed a control system for precision seed-metering device. The system adjusts the relationship between the operating speed of the seeder and the seeding motor speed, according to seed spacing and seeding rate of the seeder. Zhang et al. [22] designed an electronic control seeding system for maize. The desired rotating speed and actual rotating speed of the seed-metering device were taken as inputs, and the PID parameters were adjusted by genetic algorithm to obtain the target speed of the seed-metering device. He et al. [23] developed an electro-mechanic control system for the seed-metering unit of single seed corn seeders. To improve seeding quality and the speed of the traditional seeder, PID algorithm was used to control the rotating speed of seed-metering device. Cay et al. [24] developed an electro-mechanic control system for seed-metering unit of single seed corn seeders that combined with PID controller for corn seed-metering device. Yin et al. [25] developed a low-cost corn precision seeding control system for a corn drill. The system uses digital PID controller to compensate the seed metering speed. Chen et al. [26] designed a control system of maize precision seeding, based on a double closed loop PID fuzzy algorithm to achieve self-adjustment for parameters of PID controller.

Facing off diversified seeding objects of EDS, it must adapt to the environment of high-density seeding. Such as mung beans, red beans and other coarse grains, with seed spacing of about 100 mm. Adjusting the parameters of the PID controller [27,28] is the most critical part of this research. The controller parameter tuning problem is a multi-objective optimization problem; heuristic optimization algorithms shine as the most suitable options [29]. It is proposed to use heuristic optimization algorithms to adjust the parameters of the fuzzy PID controller. In order to further improve the ability of the current algorithm to solve the optimal solution, the genetic algorithm (GA) and particle swarm optimization algorithm (PSO) are considered to be combined for genetic screening, which can enhance the convergence ability of the PSO while giving the GA the memory ability, and thus enhancing the global optimization efficiency of the algorithm. In order to improve the auto-tuning and tamper-resistance ability of PID controller and reduce the modification and optimization difficulty of the fuzzy controller for EDS after parameter setting, a GAPSO-optimized fuzzy PID controller for EDS was studied.

Section 2 describes the process of modeling for seeding motor and specifies the control object. Section 3 introduces the establishment of an adaptive fuzzy controller in detail and determines the optimization object of heuristic optimization algorithm. Section 4 describes the GAPSO-optimized fuzzy PID controller for EDs and describes the algorithm fusion process and simulation process in detail. Section 5 is the results and discussion, mainly including the simulation experiment, the speed accuracy experiences of seed metering device, and the field experiment. Section 6 is the conclusion of the study.

## 2. Modeling for Seeding Motor

The research object was the air suction mung bean-metering device. Assuming that there was *m* seed cells in the metering disc of the seed-metering device, the time difference between two adjacent mung bean seeds falling is as follows:(1)$$\Delta t=\frac{60}{nm}$$
where Δ*t* is the falling time difference between two mung bean seeds, s; *n* is the rotating speed of the seed-metering device, r/min.

The equation of mung bean seed spacing is
(2)$$Z=277.78v\Delta t=\frac{1.67\times 10^{4}v}{nm}$$
where *Z* is mung bean seed spacing, mm; *v* is the operating speed of seeder, km/h.

The seed cells number *m* is 27 in the mung bean seed-metering disc in the selected seed-metering device, so the following equation is obtained from Equation (2).
(3)$$n=\frac{618.52v}{Z}$$

According to the technical specification for the Heilongjiang Province local standard Technical Regulations for Mechanized Production of Mung Bean (DB23/T 2767-2020), when mung beans are sown in a single row by a tractor with more than 25 horsepower, the seed spacing is 80 mm~100 mm. Take 100 mm in study and substitute the plant spacing value into Equation (3) to obtain:(4)n=6.19v

In order to establish the simulation system for designed intelligent control strategy of the seeding motor, it was necessary to establish a mathematical model of the seeding motor. JGB37-3650-1280 DC brushless reducer motor produced by ASLONG Company was selected as the seeding motor for research. The seeding motor parameters are shown in Table 1.

The mechanical characteristics equation of brushless DC motor is the same as that of separately excited DC motor. Because the fluctuation of back-EMF and torque of DC brushless motor is relatively large, the concepts of average torque and average back-EMF are used in the mechanical characteristics equation. According to the principle of brushless DC motor, the voltage–balance equation of brushless DC motor is obtained:(5)$$U_{0}=L\frac{dI}{\mathrm{dt}}+IR+C_{e}n$$
where *U*_0_ is armature voltage, V; *L* is armature inductance, H; *I* is armature current, A; *R* is armature resistance, Ω; *C_e_* is the back-EMF coefficient, V·(rad·s^−1^)^−1^; *n* is the seeding motor speed, r/min.

The torque–balance equation of brushless DC motor is as follows:(6)$$T_{e}=J\frac{dn}{\mathrm{dt}}+T_{f}$$
where *T_e_* is the electromagnetic torque, g·cm; *J* is the moment of inertia, kg·m^2^; *T_f_* is the load torque, g·cm.

According to Equations (5) and (6), the differential equation of DC brushless motor can be calculated as follows:(7)$$T_{d}T_{m}\frac{d^{2}n}{{\mathrm{dt}}^{2}}+T_{m}\frac{dn}{\mathrm{dt}}+n=\frac{1}{C_{e}}U_{0}$$
where *T_d_* is the electromagnetic time constant, ms; *T_m_* is the mechanical time constant, ms.

By Laplace transform from Equation (7), the transfer function of brushless DC motor can be obtained as follows:(8)$$G(s)=\frac{\frac{1}{C_{e}}}{T_{m}T_{d}s^{2}+T_{m}s+1}$$

The *T_d_* and *T_m_* are calculated as follows:(9)$$T_{m}=\frac{RJ}{K_{t}C_{e}}$$
(10)$$T_{d}=\frac{L}{R}$$
where *K_t_* is the torque constant, N·m·A^−1^.

The *T_d_* and *T_m_* can be calculated by Equations (9) and (10). The seeding motor parameters *T_m_* is 0.017, *T_d_* is 0.855, and *C_e_* is 0.192. They are substituted into Equation (8) to obtain the seeding motor mathematical model:(11)$$G(s)=\frac{5.21}{0.01454s^{2}+0.17s+1}$$

## 3. Establishment of Adaptive Fuzzy Controller

Compared with the general fuzzy system, the adjustment process of the error and error rate of the whole system was controlled by the same fuzzy reasoning rules. The change range of error *e* and error rate *e_c_* of the system was defined as the domain on the fuzzy set. According to the set fuzzy inference rules, the system can adaptively modify *k_p_*, *k_i_* and *k_d_*. The modified parameters could be obtained by deblurring the obtained results Δ*k_p_*, Δ*k_i_*, and Δ*k_d_*, which were the three outputs of the fuzzy controller. After substituting them into Equations (12)–(14), respectively, the three modified characteristic functions *k_p_*, *k_i_*, and *k_d_* can be obtained by calculation.
(12)$$k_{p(k)}=k_{p(k-1)}+\Delta k_{p(k)}$$
(13)$$k_{i(k)}=k_{i(k-1)}+\Delta k_{i(k)}$$
(14)$$k_{d(k)}=k_{d(k-1)}+\Delta k_{i(k)}$$
where *k_p_*_(*k*−1)_, *k_i_*_(*k*−1)_, and *k_d_*_(*k*−1)_ are the three PID characteristic functions before updating; Δ*k_p(k)_*, Δ*k_i(k)_*, and Δ*k_d(k)_* are the correction parameters of PID characteristic functions; *k_p(k)_*, *k_i(k)_*, and *k_d(k)_* represent the updated PID characteristic functions. 

Then, *k_p(k)_*, *k_i(k)_*, and *k_d(k)_* were placed into the transfer function of PID and the online automatic correction of PID parameters was completed. In order to establish a fuzzy PID controller, two input and three output structures were chosen for the fuzzy controller. Mamdani method was selected as the fuzzy inference method, and the membership function of the fuzzy controller is shown in Figure 1.

According to the membership assignment rules of each fuzzy subset and the fuzzy control model of each parameter, the PID parameters could be modified online by adopting fuzzy control rules. The fuzzy matrix table of PID modification parameters Δ*k_p_*, Δ*k_i_*, and Δ*k_d_* was designed by using fuzzy synthetic reasoning. The fuzzy inference rules were 49 kinds of conventional fuzzy inference rules. According to the membership degree assignment table of each fuzzy subset and the fuzzy control model of each parameter, the PID parameters can be modified online by using the fuzzy control rules. The fuzzy matrix table of PID controller correction parameters is designed by using fuzzy synthesis reasoning, and the designed PID correction parameters are studied Δ*k_p_*, Δ*k_i_*, and Δ*k_d_*. The 49 fuzzy inference rules are shown in Table 2. The fuzzy surfaces, corresponding to the modified parameters, are shown in Figure 2.

In the Simulink module, the quantization coefficients *k_e_* and *k_ec_* can be introduced to the inputs *e* and *e_c_* as the quantization factors of the fuzzy controller. The definition of the quantization factors *k_e_* and *k_ec_* is:(15)$$k_{e}=\frac{n_{e}}{e_{\max}},k_{ec}=\frac{n_{ec}}{ec_{\max}}$$
where *k_e_* is the quantization factor of error, *k_ec_* is the quantization factor of error change rate, *n_e_* is the fuzzy series of error, *n_ec_* is the fuzzy series of error change rate, *e*_max_ is the maximum error value, and *ec*_max_ is the error value of error change rate.

PID parameter calibration is carried out by formulating fuzzy rules of the fuzzy controller in advance. According to Equations (12)–(14), the output Δ*k_p_*, Δ*k_i_*, and Δ*k_d_* adaptively adjusts three parameters *k_p_*, *k_i_*, and *k_d_* in the PID controller; in the Simulink module, Δ*k_p_*, Δ*k_i_*, and Δ*k_d_* introduced the scaling factors *dk_p_*, *dk_i_*, and *dk_d_* as the scaling factors of the fuzzy controller. The definition of the scaling factors *dk_p_*, *dk_i_*, and *dk_d_* is:(16)$$dk_{p}=\frac{k_{p\max}}{N},dk_{i}=\frac{k_{i\max}}{N},dk_{d}=\frac{k_{d\max}}{N}$$
where *dk_p_* is the scaling factor of the proportional correction parameter; *dk_i_* is the scaling factor of the differential correction parameter; *dk_d_* is the scaling factor of the integral correction parameter; *k_p_*_max_, *k_i_*_max_, and *k_d_*_max_ are the maximum change values of the three characteristic parameters of the PID controller; and *N* is the fuzzy output series of the fuzzy controller.

It can be found clearly from Figure 1 and Figure 2, the fuzzy domain of fuzzy input and fuzzy output was taken as {−3, −2, −1, 0, 1, 2, 3}. Because there was no fixed rules for the selection of the fuzzy domain and the setting process of two inputs and three outputs, the fuzzy domain is cumbersome. The essence of fuzzy control was to change the fuzzy series under the preset fuzzy rules through fuzzy input and fuzzy output. The quantitation-scaling coefficient could be introduced into the fuzzy input and fuzzy output, respectively, and the fuzzy series could be adjusted by optimizing the coefficient, so as to achieve the best control effect.

Figure 3 shows the simulink model of fuzzy PID control system. The control system takes the theoretical speed of the seed metering device as the step input and makes the difference between the input and the feedback through the comparator to obtain the system error *e.* The system error is differentiated and the system error change rate *ec* is obtained. *e* and *ec* are the inputs of the fuzzy controller, and the three correction values of the PID controller can be obtained through the fuzzy rules. The correction value and the set value can be added to obtain the three modified characteristic functions, the corrected characteristic function is multiplied by the error, the integral characteristic function is multiplied by the error integral value, and differential characteristic function is multiplied by error differential values. The above three values are characteristic functions that affect the control effect of the PID controller, and then the PID controller outputs the control amount to the seeding motor until the target value is reached. Therefore, the introduction of scaling factors and quantization factors can directly change the correction ability of the characteristic functions of the PID controller. Quantization coefficients *k_e_* and *k_ec_* were used as quantization factors of inputs *e* and *e_c_* of the fuzzy controller. To the fuzzy outputs Δ*k_p_*, Δ*k_i_*, and Δ*k_d_*, the scaling coefficients *dk_p_*, *dk_i_*, and *dk_d_* were used as the scaling factor of the fuzzy controller. According to Z-N method, the values of three characteristic functions before the updating of the PID controller were *k_p_* = 3.5, *k_i_* = 0.9, and *k_d_* = 0.08. All quantization factors and scaling factors were set to 1. After experiment, the results of the EDS system were as follows: the maximum overshoot was 6.399%, settling time was 3.31 s, and steady-state error was −0.5287.

Fuzzy PID had obvious disadvantages, that there was no general setting rule for membership functions. The fuzzy rules and membership functions after setting the fuzzy series of the fuzzy controller could not be adjusted. The process of setting the domain and fuzzy rules was complicated; the process of resetting the domain was complex; the ideal control effect might not be obtained; and maximum overshoot, settling time, and steady-state error are difficult to achieve optimum. 

In the seeding, because of the uneven terrain, the operating speed will be often disturbed, so the settling time of the system has a great influence on the results. In the seeding, with the increase in operating time, the steady-state error accumulation of the system will become larger and larger, which is the total seeding rate deviation caused by the total stroke deviation of the seeding motor, so the system should also keep small steady-state error.

## 4. GAPSO-Optimized Fuzzy PID Controller for EDS

PSO is a swarm intelligence algorithm based on birds flocking behavior. Let N_P_ be the number of particles in PSO. A particle *P*_*i*_ has position *X_i_* and velocity *V_i_*. The fitness function is used to evaluate each particle for checking the quality of the solution. Initially, each particle is assigned with a random position and velocity values. Each particle computes its own best called *Pbest_i_* and global best called *Gbest* for every iteration [30]. To reach the global best solution, it uses its personal and global best to update the velocity *V_i_* and position *X_i_* using the following Equations (17) and (18).
(17)$$V_{i}\left(t+1\right)=\omega \times V_{i}+c_{1}\times \chi_{1}\times \left(X_{Pbest_{i}}-X_{i}\right)+c_{2}\times \chi_{2}\times \left(X_{Gbest}-X_{i}\right)$$
(18)$$X_{i}\left(t+1\right)=X_{i}+V_{i}\left(t+1\right)$$
where *w* is the inertia weight, *c*_1_, *c*_2_ are acceleration coefficients and *χ*_1_, *χ*_2_ are randomly generated values.

After getting a new updated position, the particle evaluates the fitness function and updates *Pbest_i_* as well as *Gbest* from Equations (19) and (20).
(19)$$Pbest_{i}=\{\begin{array}{c} P_{i},if(Fitness\left(P_{i}\right)<(Fitness\left(Pbest_{i}\right)\\ Pbest_{i},\ldots otherwise \end{array}$$
(20)$$Gbest=\{\begin{array}{c} P_{i},if(Fitness\left(P_{i}\right)<(Fitness\left(Gbest\right)\\ Gbest,\ldots otherwise \end{array}$$

The GA is based on the principles of natural selection. It mimics the process of survival of the fittest principle in nature by trying to maximize the fitness function. The population, which represents the optimization variable sets, is updated after each learning cycle through three evolutionary processes, i.e., selection, crossover, and mutation [31].

PSO has the memory ability, and the optimal filial-population particles can be preserved, and per-generation memory will not be forgotten because of population evolution. GA does not have memory ability, and the per-generation memory will be forgotten because of population evolution. In terms of convergence, GA has a perfect convergence analysis method, and estimates the convergence rate. However, the convergence analysis on PSO is still weakness. Although there have been a simplified version of convergence analysis, the transformation from certainty to randomness needs to be further studied. In addition, unlike GA, PSO has no evolution tools, such as crossover or mutation [31]. Both GA and PSO have disadvantages, as genetic screening combination of them can not only enhance the memory ability but also enhance the convergence ability.

The study adopts the genetic screening mechanism and proposes a GAPSO combined GA and PSO. Compared with the other two algorithms, its memory ability is enhanced with the optimal solution preservation of PSO. At the same time, it can also increase the global optimization probability. The combination steps of GA and PSO are as follows:

Step 1. Initialize the particle swarm and set the initial parameters. Such as population size, crossover probability, mutation probability, particle dimension, constraint matrix.

Step 2. Calculate the fitness value of the particle swarm using the objective function, determine the optimal position of the individual particle and particle swarm, update the optimal particle, and calculate the optimal position of individual particles (*Pbest_i_*) of the particle swarm.

Step 3. Judge whether the ultimate objective of the optimization process is achieved. The ultimate objective is usually preset as the maximum number of iterations and the fitness value of the GAPSO reaching the standard. If the ultimate objective is achieved, the iterative process will be stopped, otherwise, Step 4 will be continued.

Step 4. Update particle velocity, particle position, and optimal value. The roulette wheel selection was used to select the particle swarm. It was decided to keep three quarters of the particle swarm and eliminate a quarter of the particle swarm. The probability distribution equation of the selection probability *P_i_* is as follows:(21)$$P_{i}=\frac{f(x_{i})}{\sum_{j=1}^{n}f(x_{j})}$$
where *P_i_* is the selection probability of particle *i*; *n* is the population size; *f(x_i_)* is the single particle fitness of particle *x_i_*; *f(x_i_)* is the fitness of all particles.

Step 5. Crossover in the intermediate generation particles. The new generation of individuals is generated by adaptive arithmetic uniform crossover with the crossover probability *P_c_* of 80%. The calculation equation of *P_c_* is as follows:(22)$$P_{c}=\{\begin{array}{l} \frac{k_{1}(f_{\max}-f)}{f_{\max}-f_{avg}},f\geq f_{avg}\\ \;\;\;k_{2}\;,f<f_{avg} \end{array}$$
where *P_c_* is the crossover probability; *f_max_* is the maximum fitness of particle swarm; *f_avg_* is the average fitness of particle swarm; *f* is the maximum fitness of two crossover particles; *k*_1_ and *k*_2_ are constant.

Step 6. The non-consistent mutation is used to mutate part of all particles according to the mutation probability *P_m_*. Because it needs to conform to the natural law, *P_m_* is generally set to be less than 0.05. In order to avoid falling into local optimum. *P_m_* is 0.1 in this study. The equation for calculating *P_m_* is as follows:(23)$$P_{m}=\{\begin{array}{l} \frac{k_{3}(f_{\max}-f^{\prime })}{f_{\max}-f_{avg}},f^{\prime }\geq f_{avg}\\ \;\;\;k_{4}\;,f^{\prime }<f_{avg} \end{array}$$
where *P_m_* is the mutation probability; *k*_3_ and *k*_4_ are constant.

Step 7. Update the particle swarm. The *Pbest**_i_* is updated by calculating, and then return to Step 4. When the preset maximum iteration is reached, *Gbest* is output, and the GAPSO stops.

The process of setting control parameters by GAPSO is shown in Figure 4.

The input signal was the operating speed, and the desired rotating speed of seed-metering device was calculated from Equation (3). The seeding motor driven the seed-metering device to rotate, and the feedback value of the system was the rotor speed value of the seeding motor. The difference between the desired rotating speed of the seed-metering device and the actual rotating speed value was calculated as the error *e*, and the error change rate *e_c_* was the differential of the error *e*. The flowchart of the GAPSO-optimized fuzzy PID controller for EDS is shown in Figure 5.

The integral of time multiplied by the absolute value of error criterion (ITAE) is taken as an evaluation index in GAPSO. The optimized parameters are the initial values of three characteristic functions before updating in the PID controller, two quantization factors and three scaling factors in the fuzzy controller. Thus, the step response curve of the optimality is obtained in set iteration number.

The system designed according to the index has less oscillation and selectivity to parameters. The ITAE was used in the objective function in the study, which synthesizes the total running time and the absolute value of the system steady-state error; the smaller its value, the better the performance of the system, and the equation is:(24)$$J_{ITAE}=\int_{0}^{+\infty }t\left|e\left(t\right)\right|dt$$
where *t* is the system running time; *e(t)* is the system steady-state error.

ITAE index could evaluate the static and dynamic performance of the system, such as rapidity, accuracy, and stability. This index could achieve remarkable optimization effect on the steady-state error and settling time of the seeding motor.

## 5. Results and Discussions

### 5.1. Simulation Experiences and Results

Because there are eight parameters to be optimized, the overlong simulation running time should be avoided. Before the simulation, the iteration range of these eight parameters is set according to the previous debugging experience of the fuzzy PID controller. The iteration ranges of the three characteristic functions before updating in the PID controller are: *k_p_* ∈ [0.0001, 40], *k_i_* ∈ [0.0001, 20], *k_d_* ∈ [0.0001, 1]; the iteration ranges of the two quantization factor in the fuzzy controller input are: *k_e_* ∈ [0.0001, 10], *k_ec_* ∈ [0.0001, 10]; the iteration ranges of the three scaling factor in the fuzzy controller output are *dk_p_* ∈ [0.0001, 15], *dk_i_* ∈ [0.0001, 15], *dk_d_* ∈ [0.0001, 15].

In the coding process, the fuzzy PID controller adopts the form of two inputs and three outputs, and each input and output corresponds to seven fuzzy subsets. The fuzzy subsets from negative to positive are represented by integers from 1 to 7. The fuzzy control rule table is transformed into a two-dimensional matrix. The two-dimensional matrix is straightened and arranged to form chromosome individuals, and the length of chromosome is 7^2^ × 3 = 147.

In the GA, the crossover probability *P_c_* is set to 0.8 (80%), the mutation probability *P_m_* is set to 0.1 (10%), the chromosomes number is 1, the chromosomes length is 147, the population size *n* is set to 20, and the maximum number of iterations is 100.

In the PSO, the inertia coefficient *ω* is set to 1, the acceleration coefficients *C*_1_ and *C*_2_ are 1.4945, the population number is 10, the population size *n* is 20, and the maximum number of iterations is 100.

In the GAPSO, the crossover probability *P_c_* is set to 0.8 (80%), the mutation probability *P_m_* is set to 0.1 (10%), the chromosomes number is 1, the chromosome length is 147, the population size *n* is set to 20, the inertia coefficient *ω* is set to 1, the acceleration coefficients *C*_1_ and *C*_2_ are set to 1.4945, and the maximum number of iterations is 100.

The Matlab2020b is used as the simulation environment of experiment. The performances of GA, PSO, and GAPSO on optimizing fuzzy PID controller parameters were compared, and the advantages and disadvantages of the designed control strategy were discussed. The optimization performance data table of the three control strategies are shown in Table 3. The unit step response curves optimized by the three control strategies are shown in Figure 6. The fitness function control curves are shown in Figure 7.

According to Figure 6 and Table 3, compared with fuzzy PID control, the various performance indexes have been improved by three intelligent control strategies. Compared with the other two intelligent control strategies, GAPSO-optimized fuzzy PID controller had advantages in the maximum overshoot, settling time, and steady-state error. The maximum overshoot was 0.071%, the settling time was 0.408 s, the steady-state error was 3.0693e^−5^, and that was no oscillation after reaching steady-state. Although the control strategy of optimizing fuzzy PID parameters by GA and PSO obviously improved the performance of the system, compared with fuzzy PID, they was still a poor performance in optimizing maximum overshoot and settling time, and there was still a small amount of oscillation after reaching steady-state.

According to Figure 7 and Table 3, the GA-optimized fuzzy PID controller reached the optimum in the 41st iteration, and the fitness value was 0.7531. The PSO-optimized fuzzy PID controller reached the optimum in the 48th iteration, and the fitness value was 0.6311. The GAPSO-optimized fuzzy PID controller reached the optimum in the 65th iteration, and the fitness value was 0.5575. Compared with the other two intelligent control strategies, GAPSO needed more iterations to optimize the fuzzy PID controller. However, compared with the other two intelligent algorithms, GAPSO had the smaller fitness value optimized and the better system performance.

With the GAPSO-optimized fuzzy PID controller, at the 65th iteration, the value of *k_e_* was 6.5752. At the 65th iteration, the value of *k_ec_* was 7.5865. At the 5th iteration, the value of *dk_p_* was 0.0060. At the 5th iteration, the value of *dk_i_* was 14.5898. At the 4th iteration, the value of *dk_d_* was 0.0020. At the 5th iteration, the value of *k_p_* was 30.1425. At the 65th iteration, the value of *k_i_* was 14.2584. At the 40th iteration, the value of *k_d_* was 0.3333.

In the optimization process of quantization factors and three characteristic functions before updating, the GA-optimized fuzzy PID controller had more advantages than the other two control strategies and could faster iterate the optimal value. In the optimization process of scaling factors, the PSO-optimized fuzzy PID controller had more advantages than the other two control strategies and could reach the optimal value faster. Although the GAPSO-optimized fuzzy PID controller has many iterations, it had smaller fitness value and better control performance. Therefore, the intelligent control strategy was selected as the EDS control strategy in this study.

### 5.2. Speed Accuracy Experiences of Seed Metering Device

In order to explore the speed accuracy of the three intelligent electric-driven mung bean air suction precision seed metering device control systems studied; the actual speed of the seed metering device at the theoretical seed spacing of 100 mm; the operation speed of 3 km/h, 5 km/h, 7 km/h, and 9 km/h; the speed of 30 s after the speed of the seed metering motor is stabilized; and the average speed *N* of the seed metering device is calculated through the DT-2234A digital photo tachometer (accuracy: ±0.1 rpm) developed by Lutron company. The speed of each group is repeated five times (*N*_1_~*N*_5_), and the variation coefficient (CV), mean, standard deviation (SD), and confidence interval (CI, 99% confidence) of the actual speed of the seed metering device is calculated. In order to compare the actual control effect of three intelligent control strategies. Figure 8 is the seed metering device speed accuracy experiment. Table 4 is the verification experiment table of the speed accuracy of the seed metering device.

It can be seen from Table 4 that the mung bean EDS control system based on the GAPSO-Fuzzy-PID algorithm has a good precision effect of the seed metering device speed, good uniformity of the speed, and the variation coefficient of the seed metering device speed is less than 6.00%. When the operation speed is 9 km/h, the maximum variation coefficient of the seed metering device actual speed reaches 5.90%; when the operating speed is 7 km/h, the minimum variation coefficient of the seed metering device actual speed is 2.99%. According to the standard deviation, it can be shown that GAPSO fuzzy PID has higher speed accuracy of the seed metering device, compared with the other two intelligent control systems, especially in the high-speed seeding state. At 7 km/h, the average speed of GAPSO fuzzy PID is the closest to the theoretical value, and the difference is only 0.02. In addition, the deviation rate of the seed metering device actual speed is small, and the speed deviation rate is less than 12%. When the operation speed is 9 km/h, *N*_4_ reaches the maximum deviation speed, and the speed deviation rate is 11.31%; when the operation speed is 7 km/h, *N*_5_ reaches the minimum deviation speed, and the speed deviation rate is 0.69%. Therefore, GAPSO-Fuzzy-PID with higher rotational speed accuracy is selected for field experiments.

### 5.3. Field Experiments and Results

The field experiment was conducted on April 20th, 2021 in the Heilongjiang Province Green Grassland Pasture experimental field, zhonglv-5 mung bean hybrid was selected as experiment materials, and two-rows air-suction precision seeder reformed by Heilongjiang Bayi Agricultural University was used for the experiment. The left seeding unit was the ground wheel chain sprocket seeding (GCSS), and the right seeding unit was the EDS. The field experiment is shown in Figure 9.

The length of the plot in the experiment area was 60 m, and the preparation areas at both ends were 20 m, respectively, 120 groups of stable continuous operation data were collected in the middle section 20 m, repeated three times, and the average value of the results were calculated. Fixed parameters of the experiment: the wind pressure of the blower was 2.5 kpa, and the starting angle of seeding was 20°. According to the Chinese national standard test method of single seed (Precision) seeder (GB/T 6973-2005)**.** The qualified index (*I_q_*), multiple index (*I_mult_*), missing index (*I_miss_*), and precision index (*I_p_*) were selected as the index of seeding performance. According to the method, when the theoretical seeding spacing is 100 mm, the qualified range of seeding spacing is 50–150 mm. Under different operating speeds of 3 km/h, 6 km/h, 9 km/h, and 12 km/h, the *I_q_*, *I_mult_*, *I_miss_*, and *I_p_* between GCSS and EDS were analyzed and compared, so as to verify the superiority of the control strategy studied. The experiment statistical results are shown in Table 5.

According to Table 5, various indexes had been promoted and improved by EDS in the field experiment, compared with the GCSS. According to the technical conditions of single seed (precision) seeder (JB/T 10293-2013), which was seed spacing ≤ 100 mm under, the *I_q_* ≥ 60%, the *I_mult_* ≤ 30%, the *I_miss_* ≤ 15%, and the *I_p_* ≤ 40%.

According to Figure 10, In each operation speed, each seeding performance of EDS was better than that of GCSS, when the operating speed was 6 km/h, both the GCSS and the EDS achieved the best seeding performance, but EDS compared with the GCSS, the *I_q_* of EDS was increased by 9.16%, the *I_mult_* was increased by 1.11%, the *I_miss_* was increased by 7.88%, and the *I_p_* was increased by 8.44%. At the operating speed of 12 km/h, the *I_miss_* of three GCSS experiments were higher than 15%. It showed that under the high-speed seeding operation, the GCSS had been difficult to meet the Chinese national standard, but the *I_miss_* of EDS could meet the Chinese national standard.

The electric-driven mung bean air suction precision seed metering device control system based on GAPSO-Fuzzy-PID can effectively reduce the missing index and precision index of the original seed metering device control system, reduce the system’s false rate, make the seed spacing more uniform, and meet the qualified range of each index in the national standard. However, there was no significant increase in the multiple index. The distance distribution of GCSS under each index is shown in Figure 11. The distance distribution of intelligent EDS under each index is shown in Figure 12.

It can be seen from Figure 11 and Figure 12 that, compared with the GCSS and the intelligent EDS, the intelligent EDS can significantly reduce the missing index and the coefficient of variation in the mung bean seeding, which can make the dispersion of the actual seeded mung beans closer to the theoretical seed spacing line. This ensures that the mung bean seeds between the ridges are evenly aligned and the number is consistent, which is convenient for the follow-up field management of farmers.

### 5.4. Discussions

(1) In the process of parameter optimization of the fuzzy PID controller, GAPSO had more iterations and longer off-line optimization time. Therefore, we should consider designing a new combination method of GA and PSO. The intelligent optimization algorithm with convergence and memory function could be used to shorten the off-line optimization time and improve the optimization efficiency.

(2) With the operating speed of 12 km/h, the *I_miss_* of the EDS was obviously higher than that in medium- and low-speed operation. It did not obviously change the drawback of the increase sharply of the *I_miss_* in the high-speed seeding operation of the GCSS. After removing a number of unstable factors, such as chain-driven, this might be related to the designed defects of the air-suction seed-metering device, and it might not be able to achieve high-speed seeding under GCSS.

## 6. Conclusions

This paper has presented a GAPSO-optimized Fuzzy PID controller for electric-driven seeding. Through simulation experiments, speed accuracy experiments of seed metering device, and field experiments, it is verified that the proposed GAPSO is better than the traditional GA and PSO in the parameter tuning of the fuzzy controller; compared with the other two algorithms, the GAPSO fused by genetic screening mechanism has stronger global optimization ability under the same iteration time.; Macroscopically, the rotational speed accuracy of the seeding motor driven by the three different intelligent controllers of GAPSO-Fuzzy-PID is also higher, and the field experiments results are excellent, which provides the basic conditions for the high-density planting of crops.

## Figures and Tables

**Figure 1 sensors-22-06678-f001:**
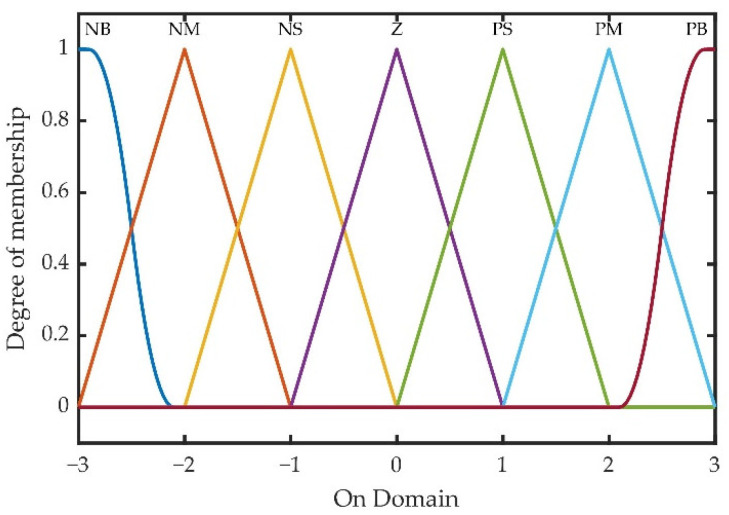
Membership functions of EDS fuzzy controller input and output.

**Figure 2 sensors-22-06678-f002:**
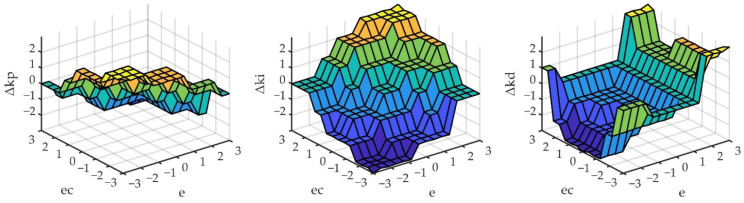
Fuzzy surface of the three correction parameters with fuzzy input.

**Figure 3 sensors-22-06678-f003:**
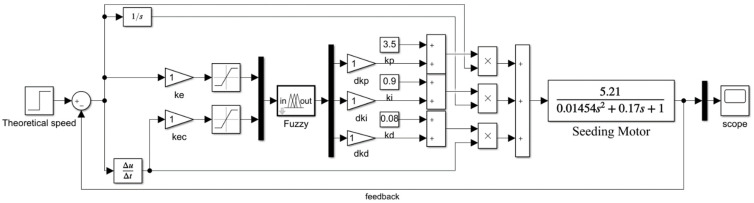
Simulink model of fuzzy PID control system with quantitation-scaling coefficients.

**Figure 4 sensors-22-06678-f004:**
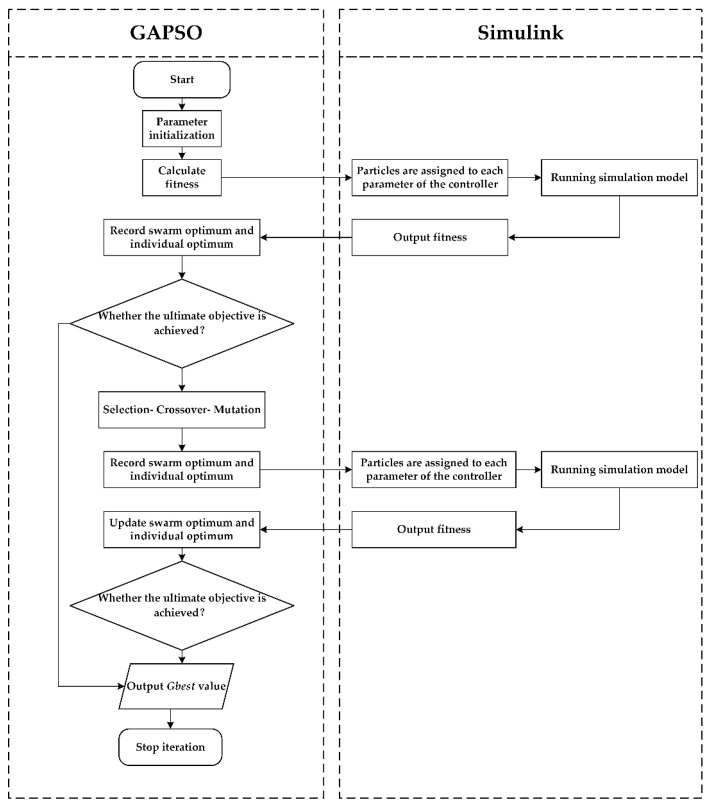
The process of setting control parameters by GAPSO.

**Figure 5 sensors-22-06678-f005:**
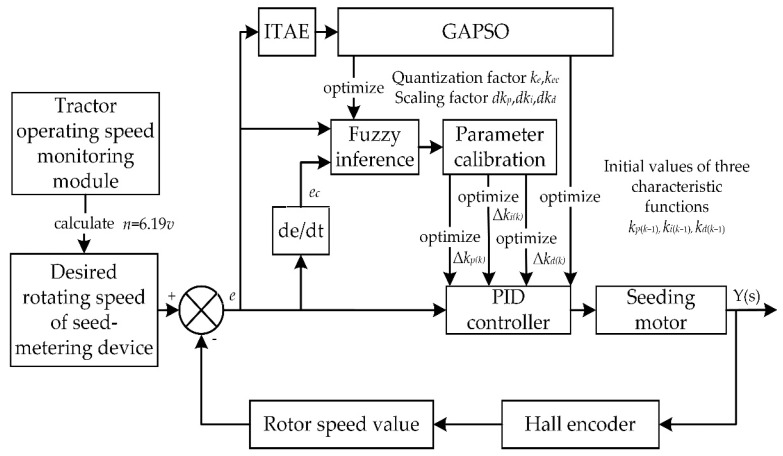
Flowchart of GAPSO-optimized fuzzy PID controller for EDS.

**Figure 6 sensors-22-06678-f006:**
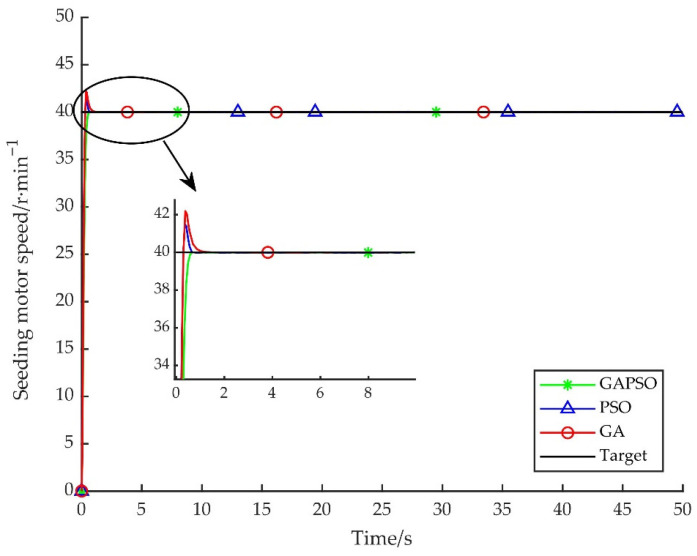
Response curves of three control strategies.

**Figure 7 sensors-22-06678-f007:**
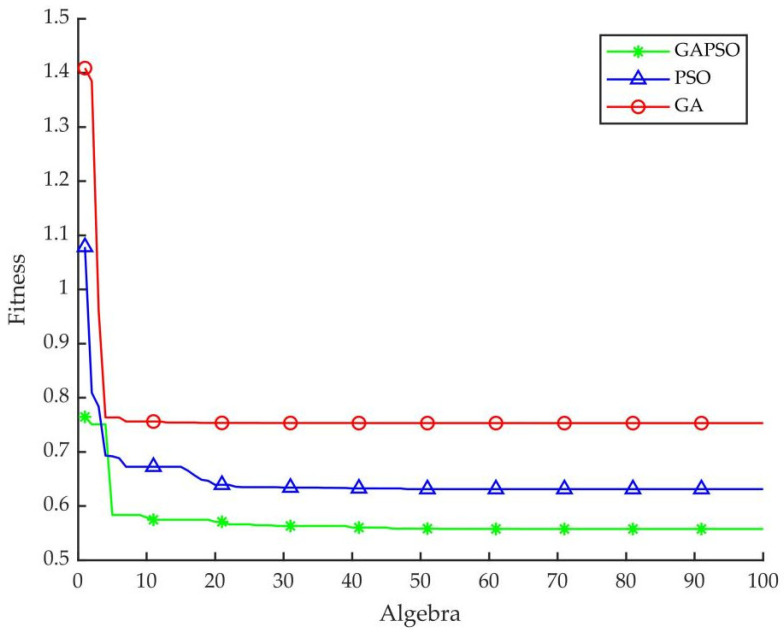
Comparison of control curves of fitness function.

**Figure 8 sensors-22-06678-f008:**
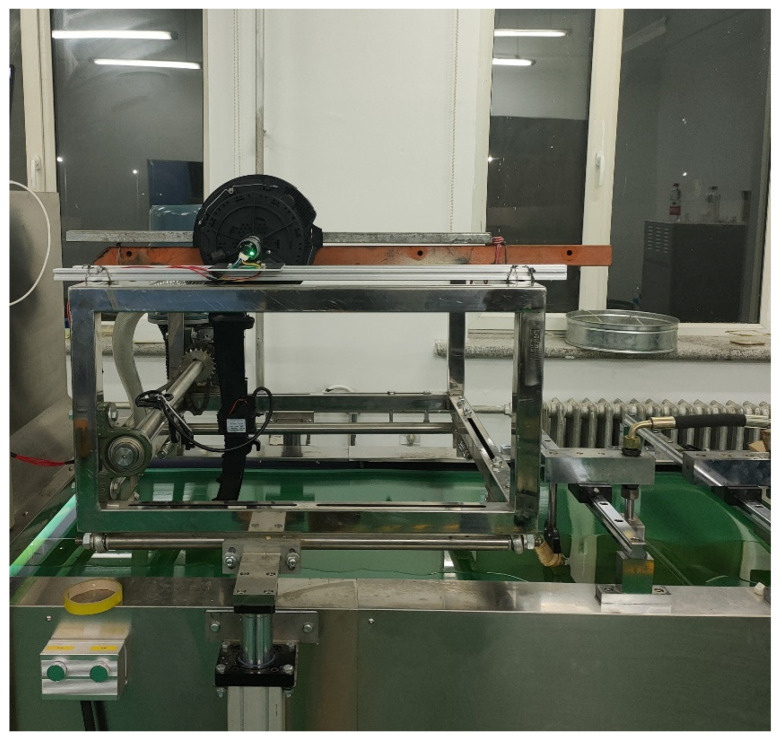
Seed metering device speed accuracy experiment.

**Figure 9 sensors-22-06678-f009:**
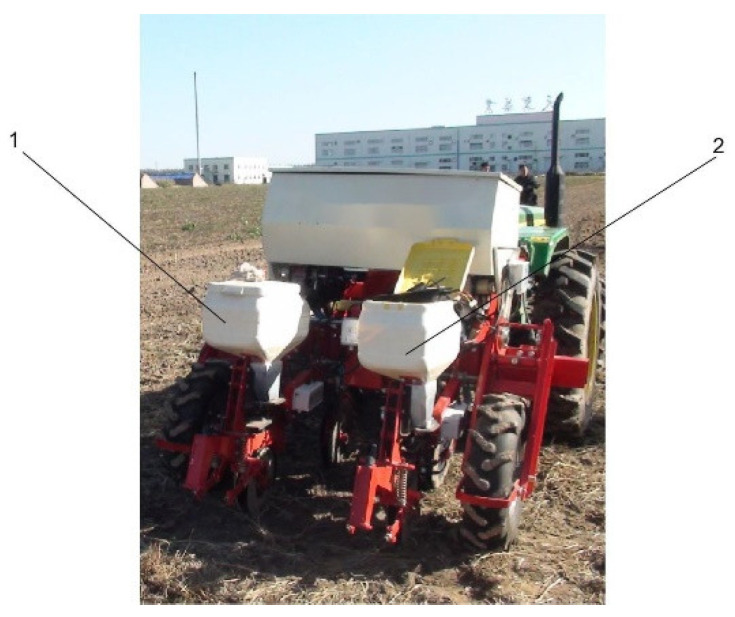
Field experiment. 1. GCSS, 2. EDS.

**Figure 10 sensors-22-06678-f010:**
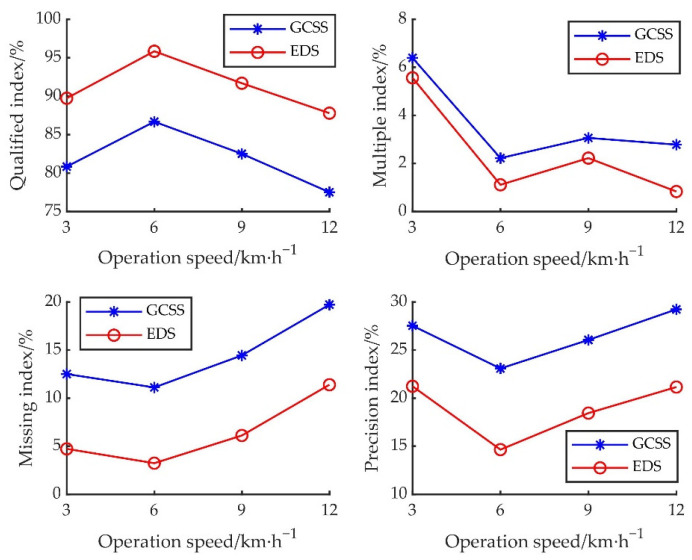
Comparison of seeding performance between GCSS and EDS.

**Figure 11 sensors-22-06678-f011:**
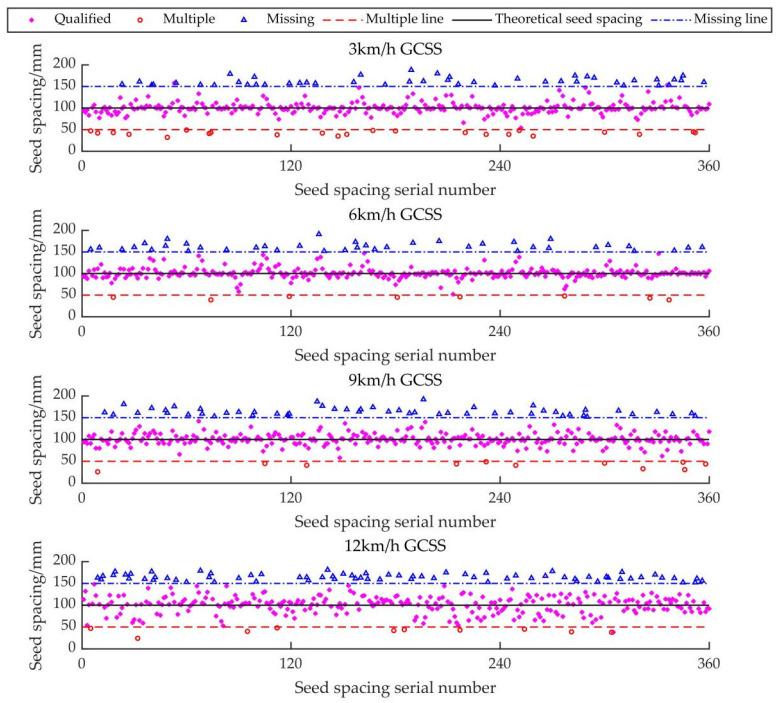
The distance distribution of GCSS under each index.

**Figure 12 sensors-22-06678-f012:**
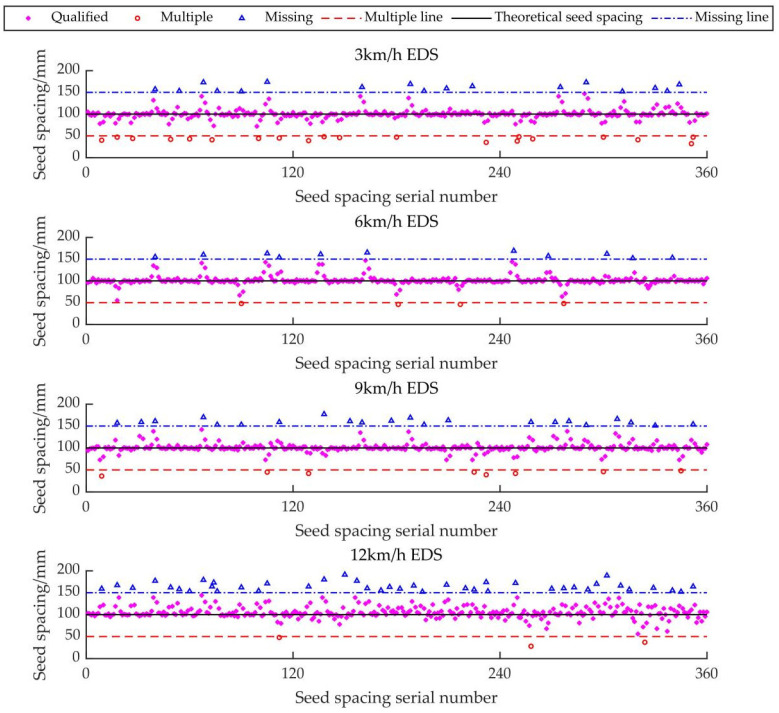
The distance distribution of intelligent EDS under each index.

**Table 1 sensors-22-06678-t001:** Seeding motor parameters.

Motor Parameters	Value	Unit
Back-EMF coefficient	0.0192	V·(rad·s^−1^)^−1^
Armature resistance	8.3	Ω
Armature inductance	7.1	H
Rotational inertia	0.0012	kg·m^2^
Torque constant	0.31	N·m·A^−1^
Working voltage	12	V
Reduction ratio	1:56	/

**Table 2 sensors-22-06678-t002:** Fuzzy inference rules table.

	NB	NM	NS	ZO	PS	PM	PB
NB	PB\NB\PS	PB\NB\PS	PM\NB\ZO	PM\NM\ZO	PS\NM\ZO	PS\ZO\PB	ZO\ZO\PB
NM	PB\NB\NS	PB\NB\NS	PM\NM\NS	PM\NM\NS	PS\NS\ZO	ZO\ZO\PS	ZO\ZO\PM
NS	PM\NM\NB	PM\NM\NB	PM\NS\NM	PS\NS\NS	ZO\ZO\ZO	NS\PS\PS	NM\PS\PM
ZO	PM\NM\NB	PS\NS\NM	PS\NS\NM	ZO\ZO\NS	NS\PS\ZO	NM\PS\PS	NM\PM\PM
PS	PS\NS\NB	PS\NS\NM	ZO\ZO\NS	NS\PS\NS	NS\PS\ZO	NM\PM\PS	NM\PM\PS
PM	ZO\ZO\NM	ZO\ZO\NS	NS\PS\NS	NM\PM\NS	NM\PM\ZO	NM\PB\PS	NB\PB\PS
PB	ZO\ZO\PS	NS\ZO\ZO	NS\PS\ZO	NM\PM\ZO	NM\PB\ZO	NB\PB\PB	NB\PB\PB

**Table 3 sensors-22-06678-t003:** Performance data sheet of simulation results.

Optimization Algorithm	Maximum Overshoot *δ*/%	Settling Time *t_s_*/s	Steady-State Error *e_ss_*	Fitness Value
GA	5.479%	0.548	0.00214	0.7531
PSO	3.748%	0.413	0.00154	0.6311
GAPSO	0.071%	0.408	3.0693 × 10^−5^	0.5575

**Table 4 sensors-22-06678-t004:** Verification experiment table of seed metering device speed accuracy.

Control Strategy	Operation Speed/km·h^−1^	Theoretical Speed/r·min^−1^	Actual Speed/r·min^−1^					CV/%	Mean	Sd	CI
			*N* _1_	*N* _2_	*N* _3_	*N* _4_	*N* _5_				
GA-Fuzzy-PID	3	18.6	17.5	19.2	20.0	16.4	20.8	9.62	18.78	1.81	±2.1
	5	31.0	28.2	33.4	31.5	29.3	34.0	8.05	31.28	2.52	±2.9
	7	43.3	40.3	45.5	41.2	44.7	44.2	5.30	43.18	2.29	±2.6
	9	55.7	51.4	62.8	59.3	49.1	50.7	11.02	54.66	6.02	±6.9
PSO-Fuzzy-PID	3	18.6	18.3	16.7	17.5	20.1	20.2	8.40	18.56	1.56	±1.8
	5	31.0	32.8	29.2	31.9	28.2	33.4	7.34	31.10	2.28	±2.6
	7	43.3	45.8	41.2	44.7	39.7	43.4	5.82	42.96	2.50	±2.9
	9	55.7	50.2	58.9	52.8	59.6	49.0	9.06	54.10	4.90	±5.6
GAPSO-Fuzzy-PID	3	18.6	17.5	18.3	19.4	16.8	18.9	5.76	18.18	1.05	±1.2
	5	31.0	28.9	30.2	31.9	32.2	30.5	4.37	30.74	1.34	±1.5
	7	43.3	44.6	42.1	43.9	41.7	44.1	2.99	43.28	1.29	±1.5
	9	55.7	49.4	56.6	57.2	54.5	52.5	5.90	54.04	3.19	±3.7

**Table 5 sensors-22-06678-t005:** Statistical results of seeding performance of two seeding methods at different operating speeds.

Operating Speed/km·h^−1^	Test NO.	GCSS				EDS			
		*I_q_*/%	*I_mult_*/%	*I_miss_*/%	*I_p_*/%	*I_q_*/%	*I_mult_*/%	*I_miss_*/%	*I_p_*/%
3	1	80.83	7.50	11.67	26.92	88.33	6.67	5.00	22.66
	2	81.67	5.83	12.50	27.32	91.67	4.17	4.17	18.85
	3	80.00	5.83	13.33	28.36	89.17	5.83	5.00	22.17
6	4	85.00	2.50	12.50	23.67	95.83	0.83	3.33	15.24
	5	87.50	1.67	10.83	23.63	96.67	1.67	1.67	13.76
	6	87.50	2.50	10.00	21.93	95.00	0.83	4.17	14.91
9	7	82.50	1.67	15.83	24.83	92.50	1.67	5.83	17.51
	8	83.33	2.50	14.16	26.07	91.67	2.50	5.83	19.10
	9	81.67	5.00	13.33	27.24	90.83	2.50	6.67	18.72
12	10	77.50	3.33	19.17	29.77	87.50	0.83	11.67	20.04
	11	76.67	2.50	20.83	29.71	88.33	0	11.67	19.18
	12	78.33	2.50	19.17	28.21	87.50	1.67	10.83	24.22

## Data Availability

Not applicable.

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
