# Peer review of "GAPSO-Optimized Fuzzy PID Controller for Electric-Driven Seeding"

_sensors, 2022, doi:10.3390/s22176678_

Round 1
Reviewer 1 Report
This work focuses on electric-driven seeding by proposing a hybrid algorithm (Genetic algorithm and particle swarm optimization). Experimental results show superior performance of the proposed algorithm. The research results also provide a reference for the parameter tuning mode of fuzzy PID controller for EDS. The proposed algorithm is important for practitioners but this paper requires some minor modifications before it is published.
1. The motivation of proposing new algorithm should be demonstrated more clearly.
2. In the bottom of Introduction, please provide the simple descriptions of Sections 2-7.
3. The main contributions of this paper should be further summarized and clearly demonstrated.
4. Introduction part needs to be extended by some of the recently published papers to show the importance of nature-inspired algorithms and applications. For example, a hybrid leader selection strategy for many-objective particle swarm optimization, a new algorithm based on PSO for multi-objective optimization.
5. There are some unclear sentences and some grammatical mistakes which make the manuscript hardly readable. Please correct them.
6. The list of references should be carefully checked to ensure consistency with between all references and their compliances with the journal policy on referencing.
7. More explanations for Figure 3 are expected as it is the most crucial part.
8. More explanations should be given to the experiments to better show the superior performance of the proposed algorithm.
Reviewer 2 Report
Every day that goes by, the relevance of seeding in today's world grows, and academic studies do too. As a result, the sudy is significant and worthwhile. However, before publishing, the authors must take into account the following suggestions/corrections. This is crucial for the study's readability and from an ethical perspective. The comments and corrections stated above are listed as follows:
1- There are numerous heuristic optimization algorithms, including the genetic algorithm, simulated annealing, and particle swarm optimization, among others. As a result, different combinations exist, like SA-GA, GA-PSO, SA-PSO, etc. Please explain the rationale for the study's choice of GA-PSO. For this case maybe the following source can be used (Optimization of fiber-reinforced laminates for a maximum fatigue life by using the particle swarm optimization. Part I, Mechanics of Composite Materials 48 (6), 705-716)
2- Similar works have been published. As a result, the importance of the study must be emphasized in depth.
3- There are important works on GAPSO, but they are not discussed in this paper. Therefore, I advise improving the GAPSO with ground-breaking studies regarding the aforementioned subject in order to boost the quality of the paper and provide in-depth information about GAPSO. For this case maybe the following source can be used (An investigation of non-linear optimization methods on composite structures under vibration and buckling loads Advances in Computational Design 5 (3), 209-231)
4- I believe the formulation contains several erroneous phrases and explanations. As a result, please double-check your formulas and make any necessary corrections.
5- The regression analysis findings are a little dubious. As a result, I recommend including values for the mean, standard deviation, in addition to COV in the experimental results. Furthermore, the confidence interval (e.g., for 95% and 99%) values of the experiments will improve the results' reliability as well as the paper's quality.
6- The paper contains an error note like follows in certain places: "Error! No such source was identified." Therefore, please provide a solution so that the relevant reference can be noticed.
7- The conclusion should be written in bullet points with only the most important findings.
8- The following studies must be evaluated and cited because the authors have used some information from them directly or indirectly, or because there are relevant research that the authors have not discussed or cited. The following are the most important studies:
- “Bold: Bio-inspired optimized leader election for multiple drones” Sensors 20 (11), 2020, article number: 3134.
- “4Nx non-isolated and non-inverting hybrid interleaved boost converter based on VLSI cell and cockroft walton voltage multiplier for renewable energy”, Proceedings of the IEEE …, 2016
9- Finally, while the work is well-written in general, it does contain some grammatical and typographical problems. Before resubmitting the manuscript, it is suggested that the authors reread it again.
Round 2
Reviewer 2 Report
It is OK to accept the paper at this time.